# HO-1 Suppression by Co-Culture-Derived IL-6 Alleviates Ferritinophagy-Dependent Oxidative Stress to Potentiate Myogenic Differentiation

**DOI:** 10.3390/cells14161234

**Published:** 2025-08-10

**Authors:** Mengyuan Zhang, Siyu Liu, Yongheng Wang, Shan Shan, Ming Cang

**Affiliations:** State Key Laboratory of Reproductive Regulation and Breeding of Grassland Livestock, School of Life Sciences, Inner Mongolia University, Hohhot 010070, China; 13674754350@163.com (M.Z.); molingyiqiu@163.com (S.L.); 32408035@mail.imu.edu.cn (Y.W.); ss18304909984@163.com (S.S.)

**Keywords:** fibro-adipogenic progenitor cells (FAPs), muscle satellite cells (MuSCs), interleukin-6 (IL-6), heme oxygenase-1 (HO-1), oxidative stress, myogenic differentiation

## Abstract

Fibro-adipogenic progenitor cells (FAPs) support muscle tissue homeostasis, regulate muscle growth, injury repair, and fibrosis, and activate muscle progenitor cell differentiation to promote regeneration. We aimed to investigate the effects of co-culturing FAPs with muscle satellite cells (MuSCs) on myogenic differentiation. Proteomic profiling of co-culture supernatants identified significant DCX, IMP2A, NUDT16L1, SLC38A2, and IL-6 upregulation. Comparative transcriptomics of mono-cultured versus co-cultured MuSCs revealed differential expression of oxidative stress-related genes (*HMOX1*, *ALOX5*, *GSTM3*, *TRPM2*, *PADI1*, and *CTSL*). Pathway enrichment analyses highlighted cell cycle regulation, TNF signaling, and ferroptosis. Gene ontology analysis of MuSCs indicated significant gene enrichment in myosin-related components. Combined transcriptomic and proteomic analyses demonstrated HO-1 downregulation at the transcriptional and translational levels, with altered pathways being predominantly related to myosin filament, muscle system process, and muscle contraction cellular components. HO-1 knockdown reduced intracellular iron accumulation in MuSCs, suppressing iron-dependent autophagy. This alleviated oxidative stress and promoted myogenic differentiation. Exogenous IL-6 (0.1 ng/mL) downregulated HO-1 expression, initiating an identical regulatory cascade, while HO-1 overexpression reversed the IL-6-mediated reduction in the expression of the autophagy markers LC3 and ATG5, suppressing myogenic enhancement. This establishes the co-culture-induced IL-6/HO-1 axis as a core regulator of iron-dependent oxidative stress and autophagy during myogenic differentiation.

## 1. Introduction

Skeletal muscle serves not only as the fundamental unit of physiological activity and metabolism but also as a primary determinant of economic output in animal husbandry. With increasing global demand for meat products, enhancing muscle growth efficiency and carcass traits (e.g., lean yield) in livestock species—including swine, cattle, and poultry—has become a critical agricultural objective. Muscle satellite cells (MuSCs), the key stem cells governing skeletal muscle regeneration and hypertrophy, exhibit myogenic differentiation capacity that directly dictates both muscle mass accrual and meat quality [1]. Consequently, elucidating the regulatory mechanisms underlying MuSC differentiation is essential for developing novel strategies to enhance meat production efficiency in farm animals.

In addition to myofibers, numerous mononucleated cells reside in the skeletal muscle, including mesenchymal cells called fibro-adipogenic progenitors (FAPs) [2]. FAPs are a conserved stromal progenitor cell population that have been identified and functionally characterized across major livestock species, including swine, cattle, and poultry (chickens) [3,4,5,6]. FAPs constitute the primary mesenchymal stromal cell population for skeletal muscle injury repair and fibrosis healing [7]. Furthermore, FAPs positively affect muscle regenerative potential and quality, support muscle growth, and provide a pro-myogenic niche during muscle regeneration [8,9,10,11]. FAPs support muscle development and repair via extracellular matrix remodeling processes, such as muscle fibrosis and adipogenesis, as well as by coordinating regulatory interactions between MuSCs and other cell types [12,13].

The microenvironment precisely regulates MuSC function. Among diverse niche factors, interleukin (IL)-6 has been extensively studied, and it exhibits paradoxical effects on myogenesis. While conventional paradigms classify IL-6 as a pro-inflammatory cytokine that may inhibit differentiation or induce muscle atrophy, emerging evidence indicates its context-dependent capacity to enhance myogenesis, particularly at moderate concentrations or within specific microenvironments [14,15]. During muscle differentiation, IL-6 promotes satellite cell differentiation and fusion through the JAK/STAT signaling pathway, particularly by activating STAT3. Therefore, STAT3 is crucial for muscle growth and regeneration [16]. However, under senescent and pathological conditions, IL-6 may directly act on muscle cells or indirectly promote muscle atrophy through interactions with other factors [17,18].

Our proteomic profiling of FAP-MuSC co-cultures demonstrated significantly elevated IL-6 secretion compared with observations in mono-cultures, with immunofluorescence confirming enhanced myogenic differentiation under co-culture conditions. Although evidence has indicated that IL-6 modulates myogenesis through oxidative stress [19], iron metabolism [20], and autophagy [21], whether these processes form an integrated regulatory cascade remains unestablished. Importantly, combined transcriptomic and proteomic analyses revealed heme oxygenase-1 (HO-1), an oxidative stress sensor, as a putative modulator of iron homeostasis and autophagic flux; however, its potential mediation of IL-6 signaling has not been explored.

In this study, we aimed to elucidate the core mechanism through which co-cultured FAPs and MuSCs promote myogenic differentiation, with a specific focus on the regulatory role of the IL-6/HO-1 signaling axis. In addition to utilizing secretome analysis of co-culture supernatants to identify key differential factors (e.g., IL-6) combined with transcriptomic and proteomic profiling of MuSCs to analyze HO-1 and associated pathway alterations, we employed HO-1 knockdown/overexpression and exogenous IL-6 perturbation experiments to define the impact of the IL-6/HO-1 axis on intracellular Fe^2+^ accumulation, reactive oxygen species (ROS) levels, autophagy markers (LC3/ATG5), and myotube formation.

## 2. Materials and Methods

### 2.1. Establishment of Co-Culture System

Cryopreserved FAPs and skeletal MuSCs of Arbas cashmere goats, stored in liquid nitrogen, were thawed. FAPs were inoculated onto a 6-well Transwell (0.4 μm) insert, and MuSCs were inoculated onto a 6-well plate. The cells were separately cultured with Dulbecco’s modified Eagle’s medium/Ham’s F-12 nutrient mixture (DMEM/F-12) containing 15% fetal bovine serum. When both cells reached 70–80% confluence, the culture media in the 6-well plate and Transwell insert were, respectively, changed to DMEM containing 2% horse serum and Opti-MEM reduced serum medium. Subsequently, the Transwell insert was placed in the 6-well plate to establish a co-culture system of the two cell types. The duration of differentiation was determined based on the state of differentiation of the MuSCs in the lower layer (approximately 3–4 days).

### 2.2. Immunofluorescence Staining

MuSCs cultured under myogenic differentiation-inducing conditions for 3–4 days were incubated with fluorophore-conjugated antibodies against MYOD1 and major histocompatibility complex (MHC). Fluorescence signals were examined under a confocal microscope and quantitatively analyzed.

### 2.3. Quantitative Reverse Transcription Polymerase Chain Reaction

Total RNA was extracted from the collected cell precipitates using the phenol extraction method. Using the PrimeScript™ FAST RT (TaKaRa, Dalian, China) reagent kit with gDNA Eraser, genomic DNA elimination was performed at 42 °C for 2 min, and reverse transcription was performed at 37 °C for 10 min and at 85 °C for 5 s to obtain cDNA. Quantitative reverse transcription polymerase chain reaction (RT-qPCR) was performed using the cDNA with the highly specific qPCR reagent TB Green^®^ Premix Ex Taq™ II (Tli RNaseH Plus) (TaKaRa, Dalian, China) under the following conditions: 95 °C for 30 s, 95 °C for 5 s, and 60 °C for 34 s for 40 cycles.

### 2.4. Immunoblotting

Total protein was extracted using a mammalian protein extraction kit (Suzhou, China). Protein samples were separated using sodium dodecyl sulfate–polyacrylamide gel electrophoresis and transferred onto a nitrocellulose membrane. After blocking with 5% skim milk powder for 60 min, the membrane was incubated with the primary antibody at 4 °C overnight and subsequently incubated with the secondary antibody. Protein expression levels were observed using the SuperKine™ West Pico PLUS Chemiluminescent Substrate (Wuhan, China) and quantified by grayscale analysis.

### 2.5. Library Construction, mRNA Sequencing, and Data Processing

The extracted mRNA was enriched using mRNA capture beads. After purification with beads, the mRNA was fragmented under a high temperature. The fragmented mRNA was then used as a template to synthesize first-strand cDNA in a reverse transcription enzyme mixture system. End repair and A-tailing were completed while synthesizing the second-strand cDNA. Next, adapters were ligated, and Hieff NGS^®^ DNA Selection Beads were used for purification to select target fragments. PCR library amplification was then performed, and detection was carried out using the Illumina NovaSeq X Plus. Differential RNA expression analysis was performed using DESeq2 software 3.21 between two different groups (and by edgeR between two samples). Genes/transcripts meeting the criteria |log2FC| > log2(1.5) and *p* < 0.05 were considered to be differentially expressed genes (DEGs).

### 2.6. Protein Sequencing and Data Processing

Protein extraction and proteomic analysis were performed by Guangzhou Genedenovo Biotechnology. Proteins were digested into peptides using trypsin following standard protocols. Nano-HPLC-MS/MS analysis was conducted on a timsTOF Pro2 mass spectrometer (Bruker Daltonics, Billerica, MA, USA) coupled to an UltiMate 3000 system (Thermo Fisher Scientific, Waltham, MA, USA), with peptides separated on a nanoflow column (400 nL/min, 50 °C) under a 60-min acetonitrile gradient. Data-independent acquisition was operated in diaPASEF mode with 22 × 40 Th isolation windows (*m*/*z* 349–1229). Raw data were processed in Spectronaut 18 (Biognosys AG, Schlieren, Switzerland) using default settings, including local normalization and MaxLFQ quantification. Proteins meeting the criteria |log2FC| > log2(1.2) and *p* < 0.05 were considered to be differentially expressed proteins (DEPs). DEPs were analyzed with Student’s *t*-test with Benjamini–Hochberg correction [22].

### 2.7. HO-1 Overexpression and Interference

An overexpression vector was synthesized by VectorBuilder and used in combination with NanoTrans™ Transfection Reagent 3000 for the transfection of MuSCs. *HO-1*-targeting small interfering RNA was synthesized by GenePharma (Shanghai, China) and used in combination with NanoTrans™ Transfection Reagent 3000 for the transfection of MuSCs. Cells separately subjected to overexpression and interference treatments were cultured to 70–80% confluence. Subsequently, the culture medium was changed to DMEM containing 2% horse serum to induce differentiation.

### 2.8. Fe^2+^ Content Measurement in Cells

MuSCs were inoculated onto round coverslips. To determine the Fe^2+^ content of viable cells, the cells were subjected to immunofluorescence staining with the RhoNox-1 divalent iron ion fluorescent probe (Cat. No.: HY-D1533, MedChemExpress, Princeton, NJ, USA), and the nuclei were counterstained with Hoechst dye. Fluorescence images were acquired under a confocal microscope, and the intensity was quantified using ImageJ 1.4 via channel-split, threshold-limited (default/dark background) measurements with background normalization.

### 2.9. Measurement of ROS Levels in Cells

Treated cells were subjected to immunofluorescence staining using the H2DCFDA (DCFH-DA) (Cat. No.: HY-D0940, MedChemExpress) probe for the measurement of ROS levels. Fluorescence images were acquired under a confocal microscope, and the intensity was quantified using ImageJ via channel-split, threshold-limited (default/dark background) measurements with background normalization.

### 2.10. Measurement of Glutathione Content in Cells

The collected precipitates of differentiated cells were washed twice with phosphate-buffered saline. After the supernatant had been discarded, the cells were resuspended in extraction buffer at three times the volume of the cell precipitate and lysed through two to three cycles of alternation between liquid nitrogen and a water bath. The supernatant was obtained and added to a 96-well plate. A glutathione (GSH) assay kit (Cat. No.: KTB1600, Abbkine, Wuhan, China) reagent was subsequently added, and the absorbance at 412 nm was measured using a microplate reader. The standard curve was plotted, and the measurement results were statistically analyzed.

### 2.11. Measurement of Malondialdehyde Content in Cells

Cells were treated using the same method as described in Section 2.8 to obtain the supernatant of lysed cells. The reaction mix of a malondialdehyde (MDA) assay kit was added to the supernatant, and the mixture was incubated in a 95 °C water bath for 30 min. After cooling on ice, the supernatant was obtained and added to a 96-well plate for the measurement of absorbance at 532 nm and 600 nm.

### 2.12. Statistical Analysis

Statistical analyses were performed using GraphPad Prism 9.5 (GraphPad Software, San Diego, CA, USA). Differences between two groups were assessed using an unpaired two-tailed Student’s *t*-test, whereas one-way ANOVA with Tukey’s post hoc test was used for multi-group comparisons. Data represent the mean ± SEM of three biologically independent experiments. Significance was determined at * *p* < 0.05, ** *p* < 0.01, and *** *p* < 0.001.

## 3. Results

To model paracrine signaling in vivo, MuSCs were co-cultured with FAPs (seeded in the upper chamber of a Transwell system) under myogenic differentiation conditions for 3–4 days. The co-culture group exhibited significant increases in myotube quantity (*p* < 0.001), length (*p* < 0.05), diameter (*p* < 0.05), fusion index (*p* < 0.05) (Figure 1A–F), and expression of myogenic differentiation markers (Figure 1G,H) compared to those in the mono-culture group.

To determine whether the enhanced myogenic differentiation under co-culture conditions was attributable to alterations in secretory factor composition, we performed data-independent acquisition proteomic analysis of conditioned medium harvested from MuSCs in the FAP-MuSC co-culture system. A total of 3805 proteins were detected, with 71 proteins significantly upregulated and 170 proteins significantly downregulated in the co-culture group (Figure 2A). Among the top 30 most significant DEPs, only 5 were upregulated whereas the remaining 25 were downregulated. The upregulated proteins included DCX, IMP2A, NUDT16L1, SLC38A2, and IL-6 (Figure 2B). Gene ontology (GO) enrichment analysis revealed that the DEPs in the culture medium were enriched in cellular components, such as mitochondrial matrix (GO: 0005759), mitochondrial part (GO: 0044429), mitochondrial inner membrane (GO: 0005743), mitochondrial protein complex (GO: 0098798), and tricarboxylic acid cycle (TCA) heteromeric enzyme complex (GO: 0045239) (Figure 2C). Kyoto Encyclopedia of Genes and Genomes (KEGG) pathway enrichment analysis indicated that the DEPs were significantly enriched in the citrate cycle (TCA cycle), oxidative phosphorylation, and pyruvate metabolism pathways (Figure 2D). These pathways are associated with programmed cell death and sarcomere assembly, which are essential processes during skeletal muscle differentiation.

Following acquisition of the proteomic profile characterizing protein alterations in conditioned medium derived from the co-culture system, we performed integrated transcriptomic and proteomic profiling of MuSCs to investigate potential regulatory mechanisms underlying their myogenic differentiation. Principal component analysis (PCA) revealed clear segregation between groups (Figure 3A). Differential expression analysis identified 347 upregulated genes and 104 downregulated genes (Figure 3B). Heatmap visualization of the top 15 significantly upregulated and downregulated DEGs revealed several candidates associated with oxidative stress pathways, including *HMOX1* (encoding a core antioxidant enzyme), *ALOX5* (implicated in lipid peroxidation), and *GSTM3* (involved in glutathione metabolism). Genes associated with oxidative stress-related functions comprised *TRPM2* (ROS-sensitive ion channel), *PADI1* (post-translational modifier under oxidative conditions), and *CTSL* (lysosomal protease activated during oxidative damage) (Figure 3C). Concurrently, we observed enrichment of multiple proliferation-related GO pathways in co-cultured MuSCs, including nuclear division (GO: 0000280) and chromosome segregation (GO: 0000070) (Figure 3D). This enrichment pattern suggests altered cell proliferation and cycle progression. KEGG pathway enrichment analysis revealed significant enrichment of cell cycle (k004110), tumor necrosis factor (TNF) signaling (k004668), and ferroptosis (k004216) pathways in the FAP-MuSC co-culture system (Figure 3E).

To functionally validate the transcriptomic landscape and directly interrogate pathway activity at the protein level, MuSCs isolated from FAP-MuSC co-cultures underwent label-free quantitative proteomics analysis. PCA revealed clear segregation between groups (Figure 4A). Proteomic quantification identified 179 significantly upregulated and 183 downregulated proteins in MuSCs derived from the co-culture system (Figure 4B). GO cellular component and molecular function enrichment analysis revealed concerted organization of myogenic structural machinery, including contractile apparatus (GO: 0030017 sarcomere, GO: 0030016 myofibril, GO: 0030018 Z disc, and GO: 0031674 I band), myosin complexes (GO: 0016460 myosin II complex, GO: 0032982 myosin filament, GO: 0016459 myosin complex, and GO: 0005859 muscle myosin complex), and contractile fiber organization (GO: 0043292 contractile fiber and GO: 0044449 contractile fiber part), alongside neuronal compartments comprising GO: 0044297 cell body, GO: 0043025 neuronal cell body, GO: 0043204 perikaryon, GO: 0030424 axon, GO: 0030425 dendrite, and GO: 0043005 neuron projection (Figure 4D). This dual specialization suggests a coordinated neuromuscular interface assembly, potentially facilitating myogenic differentiation in the co-culture system.

Based on established thresholds for transcriptomic and proteomic data, DEGs and DEPs were identified. The analysis revealed 147 DEGs without corresponding DEPs; 336 DEPs without corresponding DEGs; 18 genes/proteins exhibiting concordant differential expression at both levels; and 1 gene/protein pair showing discordant expression (Figure 5A). Nineteen genes, comprising PTGS2, HMOX1, CD34, TNNI2, DIAPH3, PLS3, LOXL4, RRAD, IQGAP3, LOC102176691, TGM1, ACTG2, HVCN1, LOC102181155, LOC102181869, PHLDA1, CASQ1, ENPP1, and KIF22, were subjected to GO enrichment analysis. The results demonstrated significant enrichment in three primary categories, namely myosin filament (cellular component), muscle system process (biological process), and muscle contraction (biological process) (Figure 5B). Notably, these three significantly enriched pathways aligned with the altered myogenic differentiation observed under co-culture conditions.

Transcriptomic analysis of MuSCs indicated that a subset of DEGs were associated with oxidative stress pathways. Among them, HO-1, a key oxidative stress regulator, was consistently downregulated at both transcriptional and protein levels (Figure 5A,B). HO-1 knockdown enhanced myogenic differentiation (Figure 6C–H), concomitant with reduced Fe^2+^ accumulation (Figure 6I) and attenuated oxidative stress in MuSCs (Figure 6J–L). KEGG enrichment analysis further linked dysregulated genes to ferroptosis, wherein HO-1 modulated ferritin-dependent autophagy to regulate oxidative stress. Experimental validation confirmed that HO-1 knockdown substantially reduced the expression of the iron storage protein ferritin light chain (FTL) and the autophagy markers LC3/ATG5 (Figure 6M).

IL-6-mediated regulation of oxidative stress has been extensively documented. Given that HO-1 is a critical modulator of oxidative stress, we investigated whether these two factors cooperatively influence oxidative stress in MuSCs under co-culture conditions. Our data demonstrated that exogenous IL-6 downregulated HO-1 expression in MuSCs (Figure 7A), concomitant with reduced iron (Figure 7B) accumulation and ROS levels (Figure 7C). However, HO-1 overexpression in the IL-6-treated system rescued the oxidative stress phenotypes (Figure 7D,E). Furthermore, protein analysis revealed that HO-1 overexpression significantly reversed the IL-6-induced downregulation of LC3 and ATG5 (Figure 7F). Similarly, the upregulation of myogenic differentiation markers promoted by IL-6 was attenuated upon HO-1 overexpression (Figure 7G–N).

## 4. Discussion

To systematically deconvolute the molecular mechanisms underlying FAP-mediated myogenic differentiation of MuSCs, we performed integrated transcriptomic and proteomic analyses. Specifically, we conducted a proteomics analysis of conditioned medium of co-cultured MuSCs (Figure 2) to identify soluble regulatory factors secreted within the FAP-MuSC co-culture system that traverse the Transwell inserts to act upon MuSCs, providing candidate molecules for mechanistic investigation. Co-culture systems are widely used to study interactions between two cell types [22,23,24]. Through our transcriptomic profiling of co-cultured MuSCs (Figure 3), we aimed to comprehensively map global gene expression changes induced by FAP co-culture, revealing regulated signaling pathways and hub genes, particularly those governing myogenesis, metabolism, and cell death. Moreover, our proteomic analysis of co-cultured MuSCs (Figure 4) validated and supplemented the transcriptomic findings at the protein level, identified post-transcriptional regulatory events, and elucidated alterations in structural/functional proteins (e.g., myofibrillar and cytoskeletal components). This tripartite omics approach enabled unbiased discovery of core molecular regulators and pathways through which FAPs orchestrate MuSC differentiation.

Our study established the IL-6/HO-1 axis as a master switch coordinating microenvironmental signaling, with metabolic reprogramming during myogenesis. Our results indicated significant changes in the content of factors, such as IL-6, IGF2R, SLC38A2, GLMP, TFPI2, and FABP3, in the culture medium of MuSCs in the co-culture group. Combined analysis of transcriptomic and proteomic data of MuSCs in the mono-culture and co-culture groups revealed that IL-6 may have mediated the significant HO-1 downregulation, thereby triggering a subsequent cascade of reactions. A previous study suggested that IL-6 upregulates its mRNA levels by acting as an autocrine factor, thereby supporting its role as an exercise-activated factor in skeletal muscle cells [25,26]. The study found that exogenous addition of 20 ng/mL IL-6 to C2C12 cells increased the intracellular IL-6 mRNA levels. Further investigation revealed that the self-stimulatory effect of IL-6 involved a Ca^2+^-dependent pathway. When intracellular blockade of Ca^2+^ was performed using a Ca^2+^ chelator, the IL-6-mediated increase in IL-6 mRNA levels was inhibited [27]. Therefore, we added an appropriate concentration of exogenous IL-6 to MuSCs to simulate the action of IL-6 elevation observed in co-culture conditions. RT-qPCR revealed that the addition of IL-6 at a concentration of 0.1 ng/mL increased the intracellular IL-6 mRNA expression level to 16 times that of the group with no IL-6 addition.

Previous research has indicated that when the secretion of pro-inflammatory factors, such as TNF-α and IL-6, is affected, HO-1 expression is typically affected. However, the direct regulatory mechanisms between IL-6 and HO-1 have not yet been reported. Wu et al. found that water buffalo horn keratin and its derived thiol-rich peptide fractions (SHPF) attenuated cellular inflammatory injury and oxidative stress through activation of the key transcription factor NRF2 and increasing HO-1 expression levels [28]. SHPF also reduced pro-inflammatory cytokine expression (IL-6, COX-2, and PGE2) and inhibited VCAM-1, ICAM-1, IL-6, and MCP-1 expression. Although the study did not determine whether a regulatory relationship exists between IL-6 and HO-1, the possibility of the existence of regulatory mechanisms between the two cannot be ruled out. Our results conclusively demonstrated that IL-6-mediated downregulation of HO-1 drives myogenesis through iron–autophagy crosstalk. However, the upstream machinery orchestrating this suppression, particularly whether IL-6 activates canonical STAT3 signaling to transcriptionally repress HO-1 [29,30] or works via non-canonical regulators (e.g., epigenetic modifiers or post-translational modifiers) [31], remains mechanistically unresolved and warrants rigorous investigation.

Differential protein enrichment analysis of co-culture-conditioned medium revealed significant pathway alterations in the mitochondrial matrix, TCA cycle, and oxidative phosphorylation, processes intimately linked to regulated cell death [32,33,34]. This association is physiologically relevant given that myogenic differentiation invariably accompanies programmed cell death in MuSCs [35]. Consistent with the transcriptomic enrichment of ferroptosis pathways, mitochondrial dynamics remodeling is strongly associated with ferroptosis execution, as has also been demonstrated by previous studies [36]. Concurrently, transcriptomic profiling identified enriched pathways associated with cell cycle progression, inflammatory responses, and ferroptosis, all critically implicated in myogenic regulation [37,38,39]. Notably, proteomic analysis directly detected enrichment of multiple myofibril-associated cellular components, while integrated transcriptome-proteomic pathway analysis demonstrated that the top three enriched pathways were exclusively skeletal muscle-related (e.g., sarcomere organization and contractile fiber assembly). This multi-modal concordance between functional assays and combined transcriptomic and proteomic analyses substantiates the experimental validity of our findings.

HO-1 causes the release of iron ions by mediating heme degradation, which increases the intracellular labile iron pool level, thereby reducing oxidative stress. A study conducted on a model of cardiomyopathy found that significant HO-1 upregulation may be involved in ferroptosis onset. A Hmox1 inhibitor significantly attenuated the onset and progression of cardiac ferroptosis in mice and exerted protective effects toward cardiac function. In contrast, the loading of hemoglobin to induce Hmox1 overexpression had the opposite effect [40]. Another study reported that mice injected with doxorubicin exhibited a decrease in hemoglobin level and an increase in the levels of iron and bilirubin (the oxidation product of biliverdin). However, a Hmox1 inhibitor or knockout of the upstream regulatory molecule NRF2 reduced iron accumulation in cardiomyocytes [41]. We showed that HO-1 downregulation reduced iron ion accumulation in MuSCs and downregulated the expression of the autophagy-related proteins LC3 and ATG5 as well as reduced cellular ROS levels. In a previous study, intervention with lipopolysaccharide increased NCOA4 (a key factor in ferritinophagy) and LC3-II (a marker of autophagy) expression and reduced ferritin expression in cells. This finding indicates that lipopolysaccharide promoted H9c2 cell ferroptosis via the activation of ferritinophagy, with LC3-II upregulation reflecting the increase in autophagic activity.

## 5. Conclusions

Integrated transcriptomic and proteomic analyses delineated a coherent signaling pathway wherein FAP co-culture with MuSCs elevated IL-6 levels in MuSCs, as confirmed by proteomic analysis of conditioned medium, resulting in significant downregulation of HO-1 at both mRNA (transcriptome) and protein (proteome) levels. Concurrent transcriptomic enrichment of ferroptosis pathways implied potential regulatory involvement. Subsequent mechanistic investigation confirmed that HO-1 suppression promotes myogenic differentiation through a coordinated reduction in iron accumulation, inhibition of ferritinophagy, and attenuation of oxidative stress (evidenced by decreased ROS/MDA and elevated GSH), collectively establishing an intracellular milieu conducive to myofibrillar assembly, a finding corroborated by enrichment of myofibril-associated components. Thus, we established the FAPs-IL-6-HO-1 axis as the core regulatory mechanism through which stromal cells enhance myogenic differentiation by suppressing iron-dependent autophagy and oxidative stress in MuSCs.

## Figures and Tables

**Figure 1 cells-14-01234-f001:**
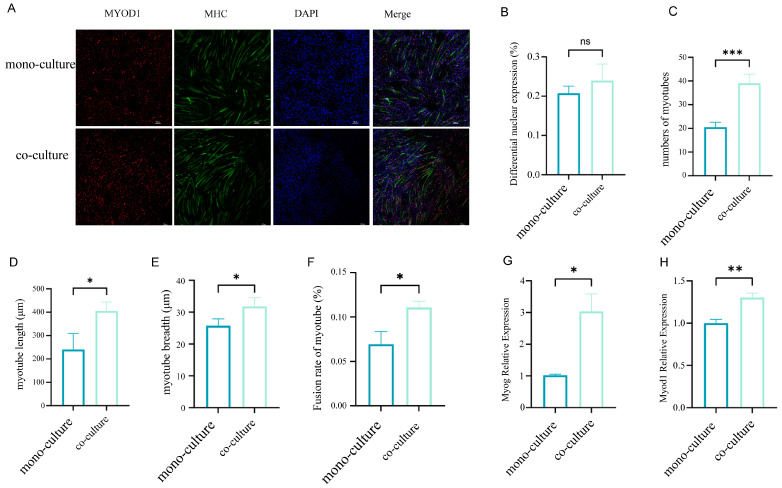
Co-culture of FAPs with MuSCs promoted myogenic differentiation of MuSCs. (**A**) Detection of myogenic differentiation of MuSCs in the mono-culture and co-culture systems by immunofluorescence staining. Quantitative fluorescence analysis of the immunofluorescence-stained images revealed the following: (**B**) The percentage of cells expressing the nuclear matrix protein MYOD did not differ significantly between the two groups; (**C**) the number of myotubes was significantly higher in the co-culture group than in the mono-culture group; (**D**) myotube length was significantly longer in the co-culture group than in the mono-culture group; (**E**) myotube breadth was significantly greater in the co-culture group than in the mono-culture group; (**F**) the co-culture group exhibited a significantly higher fusion index than the mono-culture group. (**G**,**H**) the results of the RT-qPCR showed that the expression levels of the myogenic marker genes *MYOD1* and *MYOG* were significantly higher in the co-culture group than in the mono-culture group. * *p* < 0.05; ** *p* < 0.01; *** *p* < 0.001; ns: no significant difference. (scale bar: 100 μm).

**Figure 2 cells-14-01234-f002:**
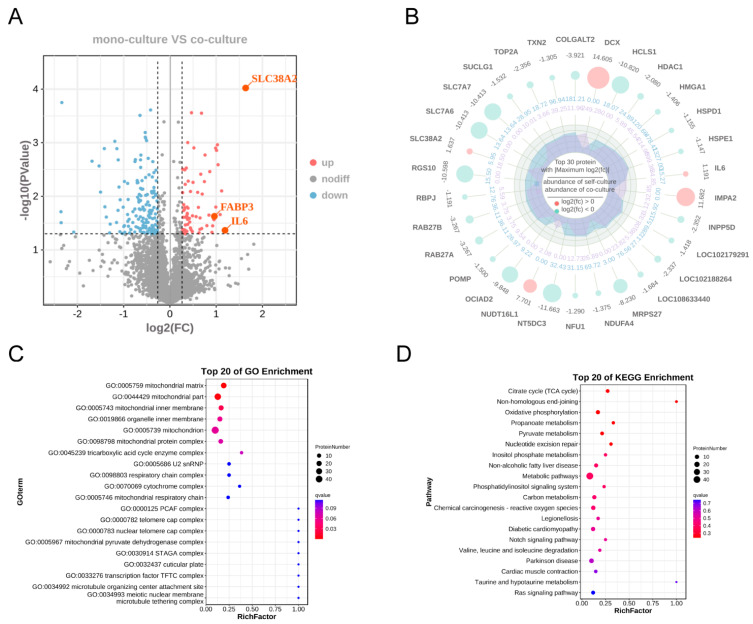
Proteomic analysis results of the supernatant of the culture medium for MuSCs in the mono-culture and co-culture groups. (**A**) Volcano plot visualizing differentially expressed proteins between the experimental groups. Proteins meeting the significance threshold (|log2FC| > log2(1.2) and *p* < 0.05) are displayed. Red points denote upregulated proteins in co-culture, blue points indicate downregulated proteins in co-culture, and gray points represent non-significant proteins. (**B**) Radar plot displaying the top 30 most significantly differentially expressed proteins between the two sample groups (ranked by ascending *p*-value). Each axis represents one protein, with radial distance from the center indicating statistical significance magnitude. (**C**) Bubble plot of gene ontology (GO) enrichment analysis. Displayed are the top 20 most significantly enriched GO terms ranked by ascending Q-value. The *y*-axis represents GO terms; the *x*-axis indicates the enrichment factor (calculated as the proportion of differentially expressed proteins annotated to the GO term relative to all proteins annotated to that term). (**D**) Bubble plot of KEGG pathway enrichment analysis. The top 20 most significantly enriched pathways, ranked by ascending Q-value, are displayed. The *y*-axis represents KEGG pathways; the *x*-axis indicates the enrichment factor (calculated as the proportion of differentially expressed proteins mapped to the pathway relative to all proteins annotated to that pathway).

**Figure 3 cells-14-01234-f003:**
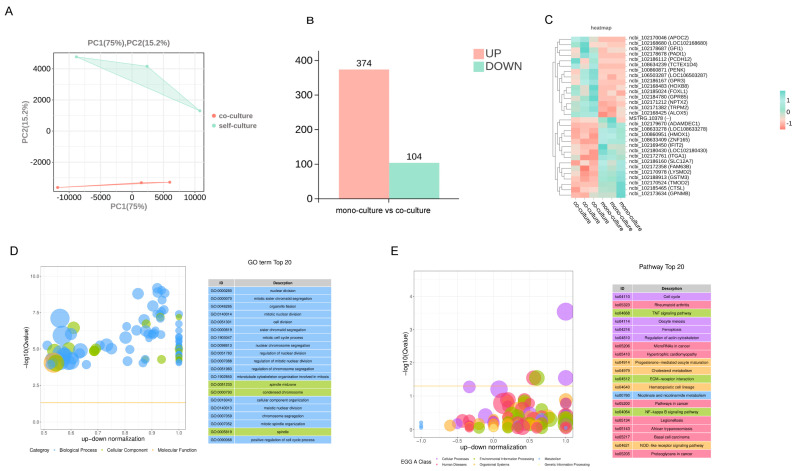
Transcriptomic analysis results of MuSCs in the mono-culture and co-culture groups. (**A**) Results of PCA performed on MuSC protein samples of the two groups. (**B**) Column chart of basic differential analysis of gene expression of MuSCs between the two groups (DESeq2,|log2FC| > log_2_ (1.5), and *p* < 0.05). (**C**) Heatmap visualization of the top 15 significantly upregulated and downregulated differentially expressed genes. (**D**) GO enrichment differential bubble plot: the *y*-axis represents −log_10_ (Q-value), while the *x*-axis represents the z-score (defined as the proportion of the difference between the number of upregulated and downregulated differentially expressed genes relative to the total number of differentially expressed genes). The yellow line indicates the Q-value = 0.05 threshold. The right panel lists the top 20 GO terms ranked by Q-value, with colors denoting distinct ontology categories. (**E**) KEGG enrichment differential bubble plot: the *y*-axis represents −log_10_ (Q-value), while the *x*-axis displays the z-score (calculated as the proportion of the difference between upregulated and downregulated differentially expressed genes relative to the total differentially expressed genes). The yellow line indicates the Q-value = 0.05 threshold. The right panel lists the top 20 pathways ranked by Q-value, with colors denoting distinct A-class categories.

**Figure 4 cells-14-01234-f004:**
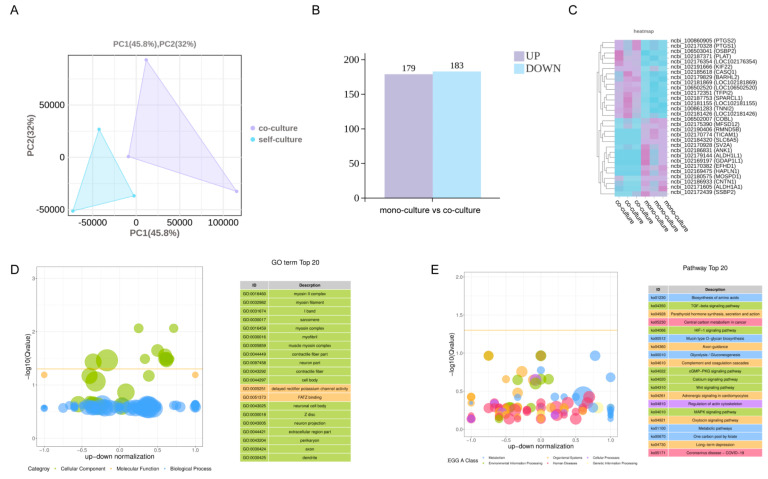
Proteomic analysis results of MuSCs in the mono-culture and co-culture groups. (**A**) Results of PCA performed on MuSC protein samples of the two groups; (**B**) Column chart of basic differential analysis of protein expression of MuSCs between the two groups (DESeq2,|log2FC|> log_2_ (1.2) and *p* < 0.05); (**C**) Heatmap visualization of the top 15 significantly upregulated and downregulated differentially expressed proteins; (**D**) GO enrichment differential bubble plot: the *y*-axis represents −log_10_(Q-value), while the *x*-axis represents the z-score (defined as the proportion of the difference between the number of upregulated and downregulated differentially expressed genes relative to the total number of differentially expressed genes). The yellow line indicates the Q-value = 0.05 threshold. The right panel lists the top 20 GO terms ranked by Q-value, with colors denoting distinct ontology categories. (**E**) KEGG enrichment differential bubble plot: the *y*-axis represents −log_10_(Q-value), while the *x*-axis displays the z-score (calculated as the proportion of the difference between upregulated and downregulated differentially expressed genes relative to the total differentially expressed genes). The yellow line indicates the Q-value = 0.05 threshold. The right panel lists the top 20 pathways ranked by Q-value, with colors denoting distinct A-class categories.

**Figure 5 cells-14-01234-f005:**
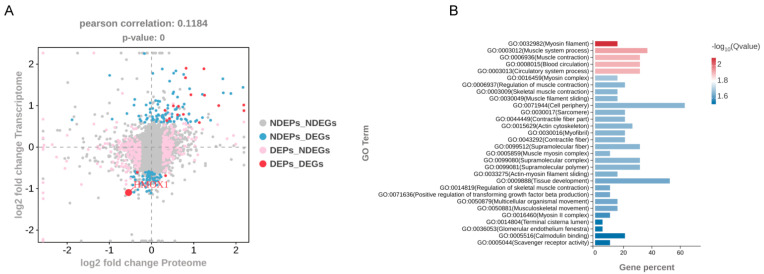
Integrated transcriptomic–proteomic analysis. (**A**) Quadrant plot displaying discordance between differentially expressed genes (DEGs) and proteins (DEPs). (**B**) GO enrichment analysis of genes exhibiting concordant differential expression at both levels (DESeq2, |log2FC| > log2(1.5), and *p* < 0.05).

**Figure 6 cells-14-01234-f006:**
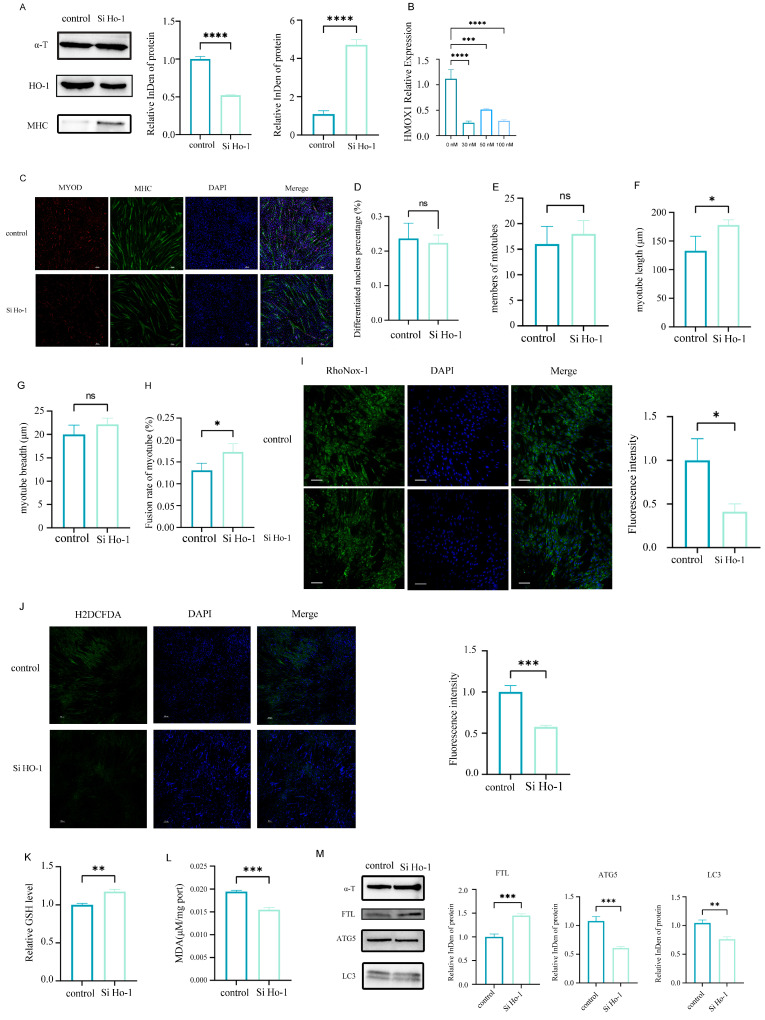
HO-1 knockdown promotes myogenic differentiation in MuSCs by attenuating oxidative stress through downregulation of ferritin-dependent autophagy proteins. (**A**) Western blot analysis of HO-1 and MHC (**left**) with corresponding quantification (**right**) after HO-1 knockdown, **** *p* < 0.0001; (**B**) Screening for the optimal concentration of si HO-1 by RT-qPCR. **** p < 0.0001, *** p < 0.001. (**C**) Immunofluorescence-stained images showing the influence of si HO-1 on the morphology of myotubes obtained from the myogenic differentiation of MuSCs, *** *p* < 0.001, **** *p* < 0.0001 (scale bar: 100 μm); (**D**–**H**) Myotube length was significantly longer in the si HO-1 group than in the control group, while the percentage of cells expressing MYOD, myotube breadth, and number of myotubes did not differ significantly between the two groups, and HO-1 knockdown significantly increased myotube fusion index, * *p* < 0.05; (**I**) Confocal microscopy showing iron ion accumulation in MuSCs stained with RhoNox-1 (**left**), with quantitative analysis of fluorescence intensity (**right**) (scale bar: 50 μm; data represent mean ± SEM of 3 independent experiments; * *p* < 0.05 vs. control); (**J**) Representative confocal images of DCFH-DA-stained cells (green; ROS signal); quantitative analysis shows significantly reduced DCF fluorescence intensity in the si HO-1 group vs. control group (data represent mean ± SEM of 3 independent experiments; *** *p* < 0.001 vs. control) (scale bar: 100 μm); (**K**) Biochemical assay measuring reduced glutathione (GSH) content. Quantitative analysis shows significantly increased GSH in the si HO-1 group vs. control group (data represent mean ± SEM of 3 independent experiments; ** *p* < 0.01 vs. control); (**L**) Thiobarbituric acid-reactive substances (TBARS) assay for malondialdehyde (MDA) content. MDA levels significantly decreased in the si HO-1 group vs. control group (data represent mean ± SEM of 3 independent experiments; *** *p* < 0.001 vs. control); (**M**) Western blot analysis of ferritin light chain (FTL) and ferritin-dependent autophagy markers. Representative immunoblots of FTL, ATG5, and LC3-II. α-tubulin served as loading control. Quantitative densitometry shows significantly elevated FTL expression in the si HO-1 group vs. control group (*** *p* < 0.001) as well as reduced ATG5 and LC3-II levels in the siHO-1 group (*** *p* < 0.001; ** *p* < 0.01). Data represent mean ± SEM of 3 independent experiments.

**Figure 7 cells-14-01234-f007:**
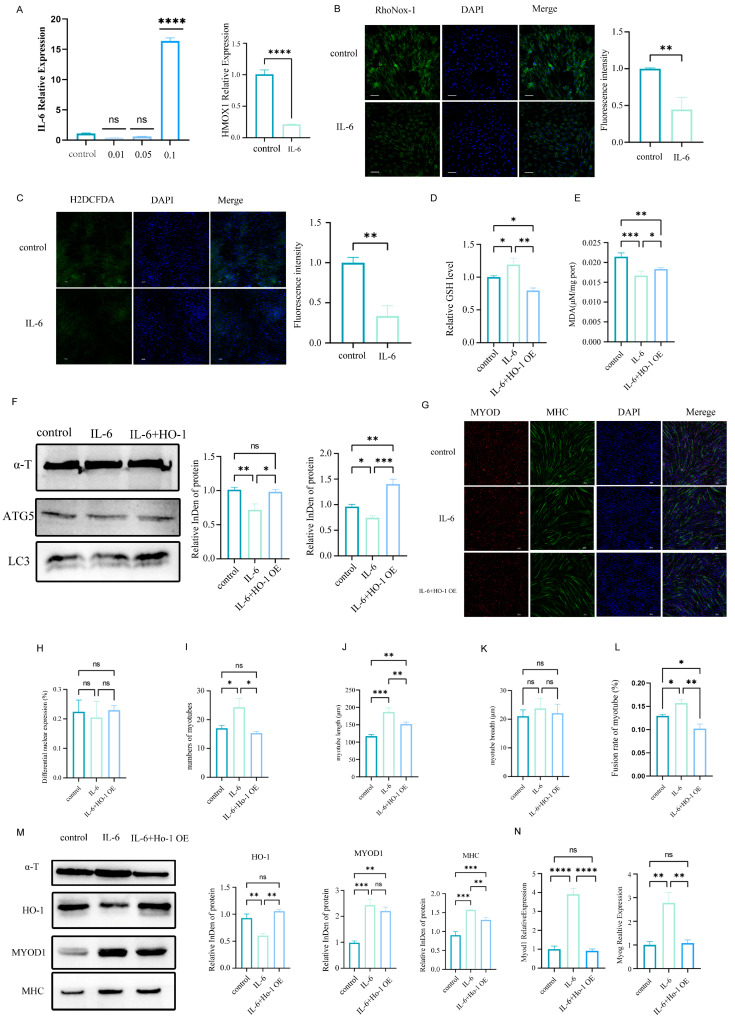
HO-1 overexpression rescues IL-6-mediated attenuation of oxidative stress and enhances myogenesis in MuSCs. (**A**) RT-qPCR analysis for optimization of IL-6 concentration (0.01–0.1 ng/mL); HO-1 expression changes in MuSCs post IL-6 treatment. HO-1 transcriptional dynamics post IL-6 treatment (0.1 ng/mL) measured by RT-qPCR (data represent mean ± SEM, n = 3), **** *p* < 0.0001, ns: Not Significant; (**B**) Confocal microscopy showing iron ion accumulation in MuSCs stained with RhoNox-1 (**left**), with quantitative analysis of fluorescence intensity (**right**) (scale bar: 50 μm; data represent mean ± SEM of 3 independent experiments; ** *p* < 0.01 vs. control); (**C**) Representative confocal images of DCFH-DA-stained cells (green; ROS signal); quantitative analysis shows significantly reduced DCF fluorescence intensity in the IL-6 group vs. control group (data represent mean ± SEM of 3 independent experiments; ** *p* < 0.01 vs. control); (**D**) HO-1 overexpression rescues IL-6-induced GSH elevation in MuSCs. Biochemical quantification of reduced glutathione (GSH). Significantly increased GSH in IL-6-treated MuSCs vs. control (* *p* < 0.05) was reversed by HO-1 overexpression in the IL-6+HO-1 OE group (** *p* < 0.01 vs. IL-6 group) (data represent mean ± SEM of 3 independent experiments); (**E**) MDA content was significantly lower in the IL-6 group vs. control group (*** *p* < 0.001) and significantly higher in the IL-6+HO-1 OE group vs. IL-6 group (* *p* < 0.05) (data represent mean ± SEM of 3 independent experiments), ** *p* < 0.01; (**F**) Western blot analysis of ATG5 and LC3 expression. Grayscale analysis shows significantly lower ATG5 (** *p* < 0.01) and LC3 (* *p* < 0.05) levels in the IL-6 group vs. control group, which were reversed in the IL-6+HO-1 OE group vs. IL-6 group (** *p* < 0.01,*** *p* < 0.001) (data represent mean ± SEM; n = 3); (**G**) immunofluorescence images of myotubes derived from MuSCs in the control, IL-6, and IL-6+HO-1 OE groups. (**H**–**L**) myotube length (*** *p* < 0.001) and number (* *p* < 0.05) were significantly greater in the IL-6 group than in the control group but reversed in the IL-6+HO-1 OE group vs. IL-6 group (** *p* < 0.01, * *p* < 0.05). No significant differences were observed among groups in MYOD-positive cell percentage or myotube breadth (data represent mean ± SEM; n = 3); (**M**) expression of HO-1 and the myogenic differentiation marker proteins MHC and MYOD1 per western blotting. Results of grayscale analysis indicated that HO-1 expression was significantly lower in the IL-6 group than in the control group, and significantly higher in the IL-6+HO-1 OE group than in the IL-6 group. MYOD1 and MHC expression was significantly higher in the IL-6 group than in the control group, and MHC expression was significantly lower in the IL-6+HO-1 OE group than in the IL-6 group (** *p* < 0.01, *** *p* < 0.001, Data represent mean ± SEM; n = 3). (**N**) Results of RT-qPCR testing showing that MYOD1 and MHC expression was significantly higher in the IL-6 group than in the control group, and *MYOD1* and *MHC* expression was significantly lower in the IL-6+HO-1 OE group than in the IL-6 group (** *p* < 0.01; **** *p* < 0.0001; data represent mean ± SEM; n = 3).

## Data Availability

The original contributions presented in this study are included in the article. Further inquiries can be directed to the corresponding authors.

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
