# Peer review of "HO-1 Suppression by Co-Culture-Derived IL-6 Alleviates Ferritinophagy-Dependent Oxidative Stress to Potentiate Myogenic Differentiation"

_cells, 2025, doi:10.3390/cells14161234_

Round 1
Reviewer 1 Report
Comments and Suggestions for Authors
The review by Zhang et al. covers an interesting topic. However, the manuscript is partly carelessly written, often repetitive and contains unnecessary phrases and repetitions that make it difficult to read and understand. Under these circumstances, it is hitherto difficult to write a complete report.
Major revision:
- The introduction is too long. For example: Listing of all responsive cells in skeletal muscle (lines 34-38) is redundant and superfluous (how are Twist2-cells relevant for this paper?) or IL-6 description over 27 lines is unnecessarily too long)
- On the other hand, one issue must be expanded in the introduction: more details to the FAP cells should be provided e.g. line 38: how often found in skeletal
- The quality of the immunohistochemical images in Figs. 1a, 5a, 5c and 5e is too low, which must be replaced by high-quality figures: Even DAPI-staining is not visible what is not acceptable.
- Draw a schematic summary of the molecular model deduced from the experiments
- The year of publication is missing in all references of the bibliography.
Minor revision:
- Line 16: which cells are in the lower layer of the co-culture system?
- Lines 18:and 285 ‚myofibril and actin cytoskeleton‘ are not understandable. It must be a physiological process.
- Lines 24, 99 and 505: Why theoretical foundation? It is experimental foundation
- Line 135: Write ‘immunoblotting’ instead of ‘Western blotting’. Western blotting is only the transfer out of the gel on the matrix.
- Lines 142, 146, 174: Do not provide information about company legal forms and trademarks. These data are not required for independent reproduction of the experiments.
- Line 184: improve typo ‘thrice’
- 181-184 Shorten the statement of significance
- Line 193: Immunofluorescence analysis is not quantitative. Provide reference for the statement in Materials/Methods
- Line 218: Why are only ‘a portion’ of the regulated proteins shown? How selected for presentation?
- Line 225: Delete ‘screened’. Superfluous.
- Line 358: It is stated that the response to exogenously added IL-6 s shown but the IL-6 response is drawn. Mention here autocrine effect, as in line 432
- Line 441: provide reference for this statement.
- 6 A1: Correct HMOX1 relative expression by deletion of M
- Literature list with numerous errors, For Example:
- References 9, 16, 25, 27, 32, 51, 52, 55 and others: Errors in journal abbreviations
- References 36, 40, 41, 42, 50: Delete ‚Supplement ….’
- Reference 31: 'Degeneration' is written in capital letters and error in the journal abbreviation
- Reference 11: Double commas
- Reference 14: Unnecessary comment in between
Author Response
Please see the attachment。

Reviewer 2 Report
Comments and Suggestions for Authors
In the following manuscript Zhang et al are presenting a manuscript describing a paracrine mechanism between ovine FAPs and satellite cells. Precisely, IL6-secreting FAPs stimulate satellite cell myogenesis via HO-1 downregulation and ferritin dependent autophagy inhibition. The overall theme is of interest and appreciated in the muscle biology field. Regrettably, the enthusiasm of this referee falls at short due to the low quality of the experiments, lack of clarity in the presentation of each panel, the poor integration of “multi-OMIC” data, the absence of validation to support the reliability of each OMIC platform as well as the absence of a conclusive finding. A detailed bulled point highlighting all referee’s concerns follows.
Figure 1 is not mentioned in the text.
Figure 1. the resolution of the immunofluorescence panel is too low. Please improve figure quality.
Please add to figure 1 the fusion index measurement.
Figure 1. Please remove the annotation (A1, A2…) and consider to canonically label panels as A, B, C… Please add in the text the proper mention to each figure panel.
Proteomics processing and data analysis is not reported in the M&M. Please detail each step as provided by reggio et 2020. The paper was the first in reporting proteome profiling of FAPs.
Proteomics analysis of MuSC culture medium. The transwell system is physically isolating two cell types. However, components in the medium can diffuse from one side to other and vice versa. Thus asserting “to investigate myogenic differentiation promotion of MuSCs by FAPs, the culture medium of MuSCs in the co-culture system was subjected to data-independent acquisition (DIA)-based proteomic analysis” is completely out of context. Authors are not profiling proteins in the “the culture medium of MuSCs”. They are profiling proteins that are released by both cell types in the transwell system. The proper control sample is not used. To elegantly interpret data, authors must also profile the secretome of FAP monoculture. Differential analysis of these three groups such as FAP monoculture, MuSC monoculture and Transwell medium will help in dissecting the source of the profiled secrete proteins.
Secretome of figure 1. “the culture medium were enriched in cellular components such as mitochondrial matrix (GO: 0005759), mitochondrial inner membrane (GO: 0005743), mitochondrial protein-con- taining complex (GO: 0098798), and tricarboxylic acid cycle heteromeric enzyme complex (GO: 0045239)” such GO terms are possibly associated to phenomenon of cell death induced by the preparation of conditioned medium. PLease specify in detail in the M&M section the procedure of preparation of conditioned media, percentage of serum, and pore sice of thranswell insert.
the dual contribution of FAPs and MuSCs to that contribution
Such observation is confirmed by the results that authors collected in figure 1A, where soluble factors released by FAPs can influence myogenesis. Please carefully rephrase the text accordingly.
Proteomic analysis of MuSC culture medium. Are the authors using FDR or p-value for establishing differences in the two-sample comparisons?
The Fold change used for the selection of significant proteins is inappropriate. A fold change > 1 or 1.5 is the standard. Alternatively provide using WB validation the reliability of the data presented.
Please highlight IL-6, SLC38A2, and FABP3 in the volcano plot to inform the reader about the magnitude of the upregulation.
Proteomics analysis of MuSC culture medium. The transwell system is physically isolating two cell types. However, components in the medium can diffuse from one side to other and vice versa. Thus asserting “to investigate myogenic differentiation promotion of MuSCs by FAPs, the culture medium of MuSCs in the co-culture system was subjected to data-independent acquisition (DIA)-based proteomic analysis” is completely out of context. Such observation is confirmed by the results that authors collected in figure 1A, where soluble factors released by FAPs can influence myogenesis. Please carefully rephrase the text accordingly.
Figure 2 must be enlarged to be readable and the resolution of each figure must be improvde. Similar outcome is expected for all figures of this manuscript
Most of the OMIC data seems to be generated without a clear experimental reason and a conclusive message from these experiments is missing, causing confusion to this referee. Please clarify better the real need of each OMIC platform, integrate data to extrapolate a conclusive biological meaning.
Proteome and transcriptome correlation must be performed to investigate correlative events.
Rephase “self-culture” with “mono-culture”.
Figure 3 and 4. In the text is not clear if these data refer to FAPs or MuSCs
Please remuvve the term Multi-OMICs from this paper author are not either performing or analyzing samples in Multi-OMIC pipeline
It is unclear from where HO-1 came out. No a single figure panel is informing the reader from the fact that this protein was found differentially expressed from transcriptome and proteome analysis.
This referee is puzzled by the siRNA experiment. No a single evidence of HO-1 silencing has been provided. Please provide expression data and protein level of HO-1 upon silencing.
It is unclear if data are presented with their standard error or standard deviation.
Attenuation of ferroptosis is not formally proven. Authors are limiting their attention to general marker of authophagy and to marker of ROS-sensitive proteins. Moreover the experiment with Erastin has not sense. To corroborate the authors’ findings are more elgant experiments must be performed including HO-1 silincing in the presence/absence of erastine. This would demonstrate a relation between HO-1 and ferroptosis.
Rhonox and other fluorescence experiments. Are these intensity data normalized? If yes, please describe in detail the procedure.
Can the model identified by authors generalized to other models, including the murine and human ones?
The following manuscripts that must be included in the background section are the following PMID 34143565, PMID: 20081841, PMID 36943314
Comments on the Quality of English Language
The language of the manuscript must be polished and improved to increase its readability.
The manuscript will likely benefit from having its language reviewed and edited by a skilled professional, preferably someone with expertise in the field and strong English language proficiency, rather than specifically requiring a "native" English speaker
Author Response
Please see the attachment。

Round 2
Reviewer 1 Report
Comments and Suggestions for Authors
By taking the reviewers' comments into account, the manuscript has become much more readable and understandable. The quality has improved significantly.
However, I must criticize two points that disappointingly have not been improved, despite the authors' announcements.
- I noted in my original comment 3 that the quality of the immunohistochemical images in Figs. 1a, 5a, 5c, and 5e is too low (now Figs 1, 6 and 7). The authors apologized for this ‘suboptimal quality’ and announced that they would replace the figures. However, the immunohistochemical images appear unchanged in black (in particular, no blue-fluorescent cellular structures are visible under DAPI labeling). Such representation is not acceptable. I would underline that the quality of the images must be significantly improved, e.g. the fluorescence signals must be enhanced using software.
- In my original comment 4, I suggested showing a schematic summary of the molecular model deduced from the experiments. The authors have included such a figure in the accompanying letter (as a graphical abstract), which is principally good. I suggest to significantly increase the font size of the molecules and integrate the drawing into the manuscript.
Author Response
1.Thank you for your continued diligence in reviewing our manuscript and for reiterating your concerns regarding the quality of the immunohistochemical images in Figures 1, 6, and 7 (previously Figs. 1a, 5a, 5c, 5e). We sincerely apologize that the adjustments made in the previous revision did not meet your expectations, and we appreciate your clear guidance on this matter.In response to your initial feedback, we carefully re-examined the original images and implemented targeted brightness adjustments while strictly adhering to two core principles:Preserving scientific accuracy: No alterations were made to the underlying data, signal distribution, or relative intensities that could affect biological interpretation.Minimal processing: Adjustments were limited to global brightness to improve visibility without introducing artifacts.
2.Thank you for your constructive suggestion regarding the schematic summary of our molecular model. We are pleased to confirm that we have now:Integrated the schematic diagram into the main text 。Significantly increased font sizes of all molecular labels and pathway elements.Maintained visual clarity while ensuring all critical interactions deduced from our experimental data remain accurately represented.
Reviewer 2 Report
Comments and Suggestions for Authors
In the present form, the manuscript has been improved significantly.
Author Response
We wish to express our profound appreciation for your exceptionally valuable guidance throughout the review process. Your insightful suggestions have been instrumental in substantially enhancing the scientific quality and clarity of our manuscript. We recognize that your expertise has elevated the rigor and impact of this work, and we are deeply grateful for your dedication to improving scholarly communication."